# Potential Treatment of Breast and Lung Cancer Using *Dicoma anomala*, an African Medicinal Plant

**DOI:** 10.3390/molecules25194435

**Published:** 2020-09-27

**Authors:** Alexander Chota, Blassan P. George, Heidi Abrahamse

**Affiliations:** Laser Research Centre, Faculty of Health Sciences, University of Johannesburg, P.O. Box 17011, Doornfontein 2028, South Africa; chotatimzy@gmail.com (A.C.); blassang@uj.ac.za (B.P.G.)

**Keywords:** *Dicoma*, cancer, bioactive compounds, medicinal plants

## Abstract

Globally, cancer has been identified as one of the leading causes of death in public health. Its etiology is based on consistent exposure to carcinogenic. Plant-derived anticancer compounds are known to be less toxic to the normal cells and are classified into acetylenic compounds, phenolics, terpenes, and phytosterols. *Dicoma anomala* is a perennial herb belonging to the family Asteraceae and is widely distributed in Sub-Saharan Africa and used in the treatment of cancer, malaria, fever, diabetes, ulcers, cold, and cough. This review aimed at highlighting the benefits of *D. anomala* in various therapeutic applications with special reference to the treatment of cancers and the mechanisms through which the plant-derived agents induce cell death.

## 1. Introduction

Cancer has been identified as a major public health problem globally. In the United states, cancer is one of the leading causes of mortality [1]. In Sub-Saharan Africa, prostate, breast, and cervical cancers are the most common with high incidence rates. The occurrence of these cancers in Sub-Saharan Africa is estimated to double up in next 20 years [2]. The cancer incidence rate is estimated to increase 50% by 2030; the burden is mostly expected to increase in low- and middle-income countries [3]. The incidence and mortality rate of cancer in low- and middle-income countries are significantly increasing, and Zambia is of no exception [4]. In South Africa, the incidence rate and treatment options of lung cancer differ between provinces because the country is diverse in terms of culture and racial groups [5]. Although the incidence rate of cancer in Africa is low when compared to other continents in the world, the mortality rates are, however, higher, which significantly reflects poor therapeutic outcomes [6].

The risk factors associated with cancers are divided into two categories based on their biological nature and modifiability. These are intrinsic and non-intrinsic risk factors. Intrinsic risk factors are simply random errors, which occur during DNA replication. Non-intrinsic risk factors are furthermore divided into endogenous and exogenous risk factors. Endogenous factors include gender, biological aging, genetic susceptibility, hormones, growth factors, and inflammation. While exogenous factors include radiation, chemical carcinogens, tumor-causing viruses, bad lifestyles such as smoking, alcohol consumption, nutrition imbalance, and lack of exercise [7]. There are different approaches that are being used in the treatment and management of cancer such as chemotherapy, radiotherapy, surgery, gene therapy, and immunotherapy [8,9]. Although these modern therapies have shown some effectiveness in the treatment of cancer, they also have adverse side effects [8]. Therefore, the research is focusing on plants and plant-derived agents to develop potent drugs to treat cancers. Many plant species have shown medicinal properties that are used in the treatment and prevention of various diseases [10]. However, it has been estimated that 50–60% of people living with cancer in the United States utilize plant-derived agents as an alternative therapy. These plants derived agents may be administered alone or simultaneously with other therapies such as chemo or radiotherapy [11].

## 2. *Dicoma* Genus

The word *Dicoma* is derived from two Greek words “*di*” (two) and “*Kome*” (tuft hair) and the word “*anomala*” is of Latin origin means irregular. *D. anomala* is widely distributed in African countries such as South Africa, Angola, Burundi, Botswana, Democratic Republic of Congo (DRC), Rwanda, Tanzania, Zambia, and Zimbabwe [12]. *D. anomala* is mostly grow in grassy and savanna areas [13]. The genus *Dicoma* has more than 10 species (Table 1).

## 3. *Dicoma anomala*

*Dicoma anomala* (*D. anomala*) is commonly known as fever or stomach bush; it is a perennial herb belongs to the family Asteraceae. It has an erect stem covered with thin hairs and bears a tuber underground [14]. As shown in Figure 1, *D. anomala* leaves are narrow, stalkless, and positioned onto the stem.

The leaves of *D. anomala* are rough and green, but most of the time, it appears grey. The flower appears like a cup with terminal heads that look white-pinkish in color [12]. It is a native plant in Sub-Saharan Africa. In South Africa, *D. anomala* is widely distributed across the country and is predominantly found in Limpompo, Gauteng, North-West, Mpumalanga, KwaZulu-Natal, Free State, and Northern Cape provinces [15]. In Africa, *D. anomala* plant has been used in the treatment of various diseases such as fever, diabetes, ulcers, cold, and cough [14]. The ethnomedicinal use of *D. anomala* are wide; its roots are extensively used in the therapy of at least 66 diseases that affect humans and animals. However, South Africa has the highest number of ethnomedicinal uses of *D. anomala* with at least 37 records for the treatment of various human illnesses [12]. Various medicinal uses of *D. anomala* species are shown in Table 2.

## 4. Role of Plants in Cancer Therapies

Plants play a very important role in the treatment of cancer. This is because many plant species possess novel chemicals that act as anticancer agents. The plant derived anticancer agents are shown to exhibit less side effects compared to many chemotherapeutic agents [16]. In past years, herbs were used as a major treatment for various cancers in different including countries in Europe and the Middle East. According to various reports from the World Health Organization, many countries use herbal medicine as an approved cancer treatment. It is estimated that only 5–15% of the herbs used in the treatment of cancer have been investigated for their anticancer properties [17], and 50,000 to 80,000 plants worldwide are being used for various medicinal purposes [18].

In order to prevent and reduce cancer incidence, researchers are trying to find alternative anticancer agents that will reduce the development of resistance caused during the chemotherapies [19]. Two-thirds of cancer therapies are derived from plant-based agents, and they are classified based on their mechanism of action. A classic example of reactive oxygen species (ROS) inducer is thymoquinone, a bioactive compound extracted from plants [17]. Apart from thymoquinone, Table 3 shows other plant-derived phytochemicals used in cancer therapies.

## 5. Mechanisms of Plant-Derived Agents Induced Cell Death 

The mechanism of action of plant-derived bioactive compounds involves inhibition of various cellular activities, and the potency of plant-derived chemical agents is dependent on the dosage [30]. Figure 2 illustrates various steps involved in tumor development and its fate. The first step involves the interaction of carcinogenic agents and reactive oxygen species (ROS) in the cell. Once inside, the carcinogens will interact with the genetic material of the cell leading to mutations. On the other hand, ROS interfere with the proteins and enzymatic activities of the cell. In step 6, the normal cell transforms into a cancerous cell, and step 7 shows the proliferation of cancer cells. The continuous proliferation of cancer cells leads to the formation of a tumor (step 8).

There are many studies conducted in the past decades. These studies reported that the in vitro activities of medicinal plants have originated from Egypt and Asian countries [18]. However, the mechanism of action of plant-derived bioactive compounds is dependent upon the quality and quantity of the phytochemicals present in them. As illustrated in Figure 2 (steps 9 and 10), the bioactive compounds are extracted from different parts of the plant and administered on the cancer cells. Once the cancer cells are treated with bioactive compounds, the cancer cells will then have two fates, as shown in steps 11 and 12, respectively. Under step 11, the cancer cells become normal after the treatment with bioactive compounds or the cells may enter in the apoptotic cell death phase by the action of phytochemicals (step 12) [16].

The possible mechanism of action of plant-derived phytochemicals are shown in Figure 3, which involves either the activation or inhibition of cellular pathways. Bioactive compounds extracted from natural sources promote apoptotic pathways, immune responses, and autophagy and inhibits different phases of cell cycle as well as the migration and invasion of cancerous cells. It is scientifically evident that scutellarein phytocompounds possess anticancer properties against a broad spectrum of cancers, including breast, colon, lung, prostate, renal, and tongue cancers. These phytocompounds induce tumor cell death through multiple cellular pathways. The molecular targets include apoptosis, cell cycle arrest, and proliferative inhibition pathways [31]. Scutellarein play an important role in tumor suppression in prostate cancer cells. They induce tumor cell death through upregulation of caspase 3, 9, G2/M-phase cell cycle arrests and Bax/Bcl-2 ratio, as shown in Figure 4 [16]. Scutellarein are also able to induce apoptotic activities in liver cancer cell line HepG2 through activation of caspase-3 enzymes and STAT3 signaling pathway [32]. This phytochemical is also able to suppress the invasion and migration of human hepatocellular carcinomas via inhibition of Akt-STAT3/Girdin activities [33]. This anticancer agent suppresses tumor growth and induces apoptotic activities against human colorectal cancer by regulating p53 [34]. In breast cancer, scutellarein inhibits tumor proliferation and inversion via upregulation of Hippo/Yap signaling pathways [35].

Cirsimaritin is a flavonoid with diversified pharmacological activities including antioxidant, anti-inflammatory, antimicrobial, anticancer, and enzyme inhibitory activities. This bioactive compound belongs to a class known as 7-O-methylated flavonoid [36]. Its anticancer activities have been explored using different cancer cell lines, such as breast, lung, and gallbladder [37]. This anticancer compound plays a vital role in angiogenesis inhibition through the downregulation of p-Akt, p-ERK, and VEGF in MDA-MB-231 breast cancer cells [38]. In human gallbladder carcinoma GBC-SD cells, cirsimaritin inhibits the growth of cancerous cells through mitochondrial apoptosis. Cirsimaritin triggers endoplasmic reticulum stress by forming ROS, and it downregulates the phosphorylation of Akt [39].

β-farnesene is one of the essential sesquiterpenes used in the treatment of different types of cancer including breast, lung, and prostate [40,41,42]. This organic compound is suppressing tumor proliferation through tumor apoptotic pathways. In tumor apoptotic pathway, ROS are formed and induce mitochondrial damage leading to cytochrome c release. The released cytochrome c together with Apaf 1 creates apoptosome that activates caspase cascade to induce cell death [43].

β-sitosterol is an important plant-derived bioactive compound used for the treatment of different medical conditions. It has diversified medicinal benefits such as anti-inflammatory, antioxidant, antidiabetic, antifertility, antimicrobial, immunomodulatory, and anticancer properties [44]. β-sitosterol inhibits tumor cell proliferation and activates the apoptotic pathway of various cancer cell lines. Cancer cell lines through which β-sitosterol exerts its anticancer activities include, breast, colon, lung, gastric, and prostate cancers [45]. In A549 cells, β-sitosterol target the enzyme Trx/Trx1 reductase to induce apoptosis through ROS and p53 activation [46]. β-sitosterol anticancer activities have been explored and induce endoreduplication in HL60 and U937 cell lines via PI3K/Akt and Bcl-2 pathways. In pancreatic cancer cell lines, β-sitosterol inhibits tumor cell growth, inducing G0/G1-phase cell cycle arrest, and apoptotic activities, downregulates NF-kB activities, upregulates the expression of Bax, and downregulates the expression of Bcl-2 protein [47].

A-Humulene is a naturally occurring bioactive compound isolated from *Eupatorium odoratum* L and has been explored for its anticancer properties. This compound uses various pathways to induce tumor cell death. In vitro effects of α-Humulene include increased production of ROS and inhibition of Akt activation [42,48].

## 6. Major Anticancer Compounds of *Dicoma*

There are various compounds that are extracted from *Dicoma* species. Out of all the *Dicoma* species, *D. schinzii*, *D. capensis*, *D. anomala*, and *D. zeyheri* have been investigated for medicinal use and were classified based on their phytochemical composition. However, only one species (*D. schinzii*) out of the four exhibited similarities in terms of bioactive compound composition. These compounds are classified into different groups: acetylenic compounds, phenolic acids, flavonoids, sesquiterpenes, triterpenes, and phytosterols. These natural compounds derived from plants are seen to be non-toxic to non-cancerous cells or normal cells, and these compounds are used in the treatment of different cancers such as breast, ovarian, prostate, and kidney [12,49].

The roots and leaves of *Dicoma anomala* are extensively used in the treatment of various diseases in Africa (Table 2). *D. anomala* extracts exhibit anticancer properties that are widely used in the treatment of breast and lung cancers. The polyphenolic compounds from *Dicoma* species such as flavonoids, sesquiterpenes, phytosterols, and triterpenes, as shown in Table 4, are considered to exhibit anticancer properties [12].

## 7. *D. Anomala* and Pharmacological Studies 

*Dicoma anomala* is a perennial herb that is widely distributed in Sub-Saharan Africa. Its aerial parts have been analyzed for the presence of phytochemical compounds [15]. *D. anomala* has been studied for its pharmacological potential in the treatment of various diseases. *D. anomala* extracts are reported for their antibacterial, anti-inflammatory, antiviral, antioxidant, anticancer, and antiplasmodial properties [58].

Vlietinck et al. (1995) evaluated the antibacterial properties of *D. anomala* root extract against various microorganisms including *Microsporum canis*, *Candida albicans*, *Staphylococcus aureus*, and *Trichophyton mentagrophytes* using agar well diffusion and dilution methods. Furthermore, they investigated the effects of *D*. *anomala* root extract on viruses such as coxsackie, herpes simplex, measles, semliki forest, and poliomyelitis through viral titer reduction [12].

Becker et al. [59] evaluated in vitro antiplasmodial activities of *D. anomala* subsp. *Gerrardii* extract. Phytochemicals such as eudesmanolide and dehydrobrachylaenolide were isolated and used in the in vitro parasite lactate dehydrogenase assay against plasmodium falciparum (p. falciparum). Dehydrobrachylaenolide demonstrated anti-malarial activities against plasmodium falciparum. Furthermore, *D. anomala* has anticancer properties that have motivated the researchers to focus on the field of biomedicine. Shafiq et al. [53] evaluated the in vitro antiproliferative effects of silver nanoparticles synthesized from the roots of *D. anomala* Sond. against MCF-7 breast cancer cells and NF54 parasitic strains. The study revealed that silver nanoparticles conjugated sesquiterpene have antiparasitic activities against NF54 *p. falciparum* strain, and it exhibited anticancer properties by inducing oxidative damage in breast cancer cells. Asita and colleagues evaluated the modulation of *D. anomala* and Cyclophosphamide (CP) in mutagen induced genotoxicity [60].

Steenkamp and Gouws in 2006 evaluated the cytotoxic properties of six plant extracts used in cancer therapy in South Africa. In the study, aqueous extract from *Dicoma capensis* demonstrated anticancer properties against three different breast cancer cell lines such as MCF-7, MDA-MB-231, and MCF-12A. The aqueous root extracts from *Dicoma anomala* were investigated for the possible postprandial extenuation in hyperglycaemia and how it modulates the activities of carbohydrate metabolising enzymes [14]. Balogun and Ashafa evaluated the activities of *D. anomala* and *Gazania krebsiana* used by Basotho for the treatment of various diseases [61].

## 8. Cancer Stem Cell Treatment

Stem cells are unspecialized body cells that have the ability to replicate and make well-differentiated specialized cells. These cells are isolated from inner mass of a cell, which is 5–8 days old. The use of stem cells in a clinical setup is restricted due to ethical, legal, and religious controversies [62]. Stem cells can be obtained from two main sources: adult and embryonic cells such as blood and placenta [62,63]. Stem cell therapy has various applications in a clinical setup including, cardiac, brain, skin, liver, and cancers [64]. The mechanism through which *D. anomala* regulates the cancer stem cell activities is not clear, but various studies show effective regulation of cancer stem cell signaling pathways. However, many plant-derived anticancer compounds target the surface targeting antibodies of cancer stem cell (CSC), such as those found on breast cancer stem cells (anti-CD133, CD44, and anti-EpCAM). These anticancer compounds may also target the ABC cassette, immune-evasion antibodies, and cytokines [65].

## 9. Breast Cancer

According to the global burden of disease (GBD) report of 2015, cancer is the second leading cause of death globally [66]. The global statistics show that 18.1 million new cases of breast cancer and 9.6 million deaths were reported in 2018. Breast cancer accounts up to 38.5% of female cancers [67]. It was also estimated that globally in next 5 years, the prevalence of breast cancer will be around 43.8 million [68]. Despite the incidence rate being low, the mortality rate of breast cancer among black African people remains higher than 40% [69].

Cancer is mainly classified based on origin. Breast cancer is named after the breast tissue with erratic growth and proliferation of cells. The breast is mainly composed of two different vital tissues: stromal and glandular tissues. The stromal tissue also known as supporting tissue, which includes fatty and fibrous connective tissues, while the glandular tissues are made up of lobules and ducts [70].

There are many types of cancer that develop in various parts of the breast. Most of them develop from the cells lining the ducts and the lobules [70]. These types of cancer are in situ or invasive. The two main types of breast carcinoma in situ are ductal carcinoma and lobular carcinoma. Ductal carcinoma in situ is known to be the precursor of invasive cancer, while lobular carcinoma in situ is a benign condition [71].

There are various risk factors that are associated with development of breast cancer. These risk factors including aging, family history, low parity, estrogen, and life styles such as alcohol abuse can lead to the development of breast cancer. There have been advancements in clinical and breast cancer related theoretical studies. However, these studies have helped in the development of breast cancer preventive measures. Currently, breast cancer preventive measures include screening, biological prevention, and chemoprevention [72].

The treatment and management of breast cancer is dependent on the stage and type of tumor. The common treatment modalities for breast cancer include chemotherapy, surgery, human epidermal growth factor receptor 2 (*HER-2*) directed therapy, endocrine therapy, and radiotherapy [73]. The introduction of natural plant-derived anticancer compounds has improved in the treatment of various cancers including breast cancer. Several reviews have highlighted the benefits of naturally derived phytochemicals as compared to the synthetic compounds. Some of the common natural bioactive phytochemicals used in anticancer therapies are vitamin E, hydroxytyrosol, resveratrol, etc. Apigenin is a phytochemical extracted from parsley vegetables are known to demonstrate cytotoxic activities against colon and breast cancer cell-lines [10].

## 10. Lung Cancer

Lung cancer is the most common neoplasm that occurs among men and women in most countries. The GLOBOCAN 2012 estimated a total of 1,242,000 new cases in men and 583,000 in women. Lung cancer is histologically classified into two classes. These classes include small-cell and non-small-cell lung cancers (NSCLC). In the United States, it was approximated that 200,000 persons were diagnosed with lung cancer in 2010 and 160,000 deaths were reported. It is the main cause of mortality in both men and women. Approximately 27% of cancer deaths, which was reported in the USA during 2015 and 20% deaths reported from European Union in 2016 were due to lung cancer [74]. Tobacco usage is the well-known risk factor that accounts for 80% to 90% of lung cancer development [75]. There are various risk factors that are associated with the development of lung cancer. The major risk factors include age and cigarette smoking. Cigarette smoking increased dramatically in the United States of America and the European countries [76].

Physical examinations, radiological imaging, and biopsy tests are performed in non-small-cell lung cancer, and the staging is based on the outcome of these investigations. Surgery can be performed in order to determine the pathological stage of cancer by direct examination of biopsies [77]. Most medical centers and hospitals have adopted the TNM staging system used in the staging of cancer where T is the size of the tumor; N is the number of lymph nodes that are infected, while M refers to the level of metastasis [78].

Therapeutic approaches to be employed in the treatment of lung cancer are dependent on the stage of cancer. Currently lung cancer is treated by chemotherapy, immunotherapy, targeted therapy, and radiotherapy. Although immunotherapy has made tremendous progress in the treatment and management of lung cancer, chemotherapy is the current standard treatment to be employed in the first and second line in the management of small-cell lung cancer (SCLC) [79]. Pre-clinical trials of 6-Shogaol, a bioactive compound extracted from ginger, have shown effectiveness in the treatment of non-small-cell lung cancer (NSCLC). In an experimental model involving a nude mouse, 6-Shogaol inhibited the growth of lung cancer cells. The growth inhibition of NSCLC was significantly associated with the decreased proliferation and increased apoptosis induction [27].

## 11. Colorectal Cancer

Colorectal cancer is the second cause of cancer-related death in the United States. Globally, colorectal cancer is the third leading cause of death in both men and females. According to the GLOBOCAN report of 2018, colorectal cancer is the fourth common diagnosed type of cancer [80]. The increasing incidence rate of colorectal cancer in developed countries can be related to risk factors such as dietary, smoking, obesity, and age [81]. Surgery remains the primary treatment modality if early diagnosed [82]. The introduction of plant-derived phytochemicals plays an important role in the treatment and management of colorectal cancer. Phytochemicals of natural origin inhibit colorectal tumor growth through pathways such as phosphoinositide 3-kinase (PI3 kinase), STAT 3, and Wnt signaling pathway [10].

## 12. Prostate Cancer

Prostate cancer is one of the most commonly diagnosed cancers found in men after lung cancer. Despite advancements in the treatment of cancer and research, prostate cancer deaths had risen to more than 300,000 deaths worldwide. The risk factors related to one developing prostate cancer include age and family history [83]. When compared to other countries, prostate cancer is more diagnosed in American and European men [84]. Therapeutic approaches to be employed in the treatment prostate cancer include chemotherapy, prostatectomy, and radiation therapy [85].

## 13. Scope and Importance of *Dicoma* in Anticancer Research

Plants are reservoirs of various phytochemicals [68]. Tumor recurrence and side effects elicited by chemotherapy have reduced the efficacy of most of the anticancer agents. Phytochemical constituents from *D. anomala* play a very important role in the development of new potent anticancer agents. Apart from being widely used in therapeutic purposes, *D. anomala* extracts are also used in industries to manufacture drugs, herbicides, and cosmetics [86]. Combination of alkaloid derivatives with other therapeutic agents proved to be more effective in the treatment of different types of cancer [87].

## 14. Combination Therapies Using Plant-Derived Phytochemicals

Cancer is a global health problem that leads to increased morbidity and mortality. Therefore, various research projects are focusing on the development of effective therapeutic approaches that will prolong human lives. Combination of cancer therapies is aimed to reduce the possibility of drug resistance during chemotherapy [88]. Since 4500 BC, plant extracts have been used in traditional practices by Indian and Chinese people. Advances in analytical chemistry have improved in the investigation of plant-derived extracts for potential medicinal components [27]. Combined therapy is the new direction taken to fight against cancer. It has a high efficacy when compared to monotherapy and provides an improved treatment efficacy and with minimal adverse effects [89].

## 15. Conclusions and Future Perspectives

Due to an increased resistance and adverse side effects exhibited by radiotherapy and chemotherapy, plant-derived bioactive compounds are considered as a potential source of anticancer agents with less side effects and cost effectiveness. Not only in cancer treatment, phytochemicals of plant origin are used by most countries in the treatment of various medical conditions. Bioactive phytochemicals are known to have anticancer, antibacterial, anti-inflammatory, antiviral, antioxidants, and antiplasmodial properties. Therefore, more efforts are being made by researchers to explore different plant extracts and identify the potent active principle compounds. Present research outcomes create a baseline for the validation and standardization of plant-derived drugs. *D. anomala*, one of the little-known African medicinal plants, has diversified pharmacological and phytochemical properties that need to be subjected to a detailed evaluation with special reference to cancer.

Due to the diversified pharmacological properties of plant-derived bioactive compounds, many researchers are focusing on developing plant-derived products for cancer treatments. Stem cell therapy by using plant-derived bioactive compounds and targeted biomarker development will replace the current synthetic ones. There are many anticancer phytochemicals without a clear mechanism of action, hence many of these compounds have to be subjected to clinical trials. Clinical trials are important, because they help in validating efficacies and adverse effects associated with those compounds. Combining traditional medicinal practices with modern treatments is one of the suggested approaches in the treatment and management of various types cancer for improved effectiveness.

## Figures and Tables

**Figure 1 molecules-25-04435-f001:**
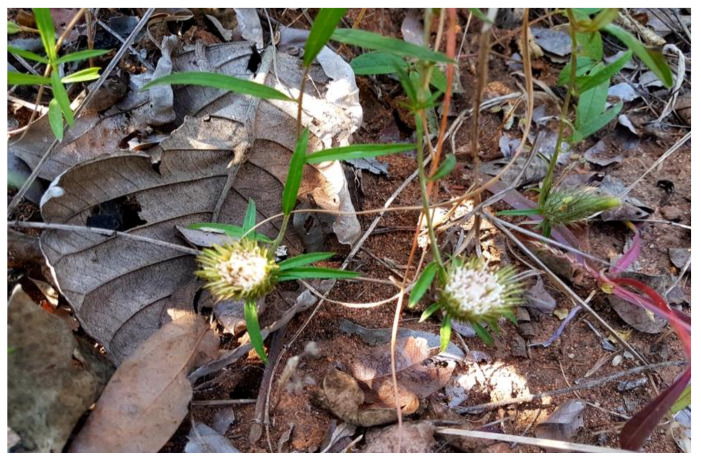
*Dicoma anomala* habitat. *Dicoma anomala* plant from eastern province of Zambia.

**Figure 2 molecules-25-04435-f002:**
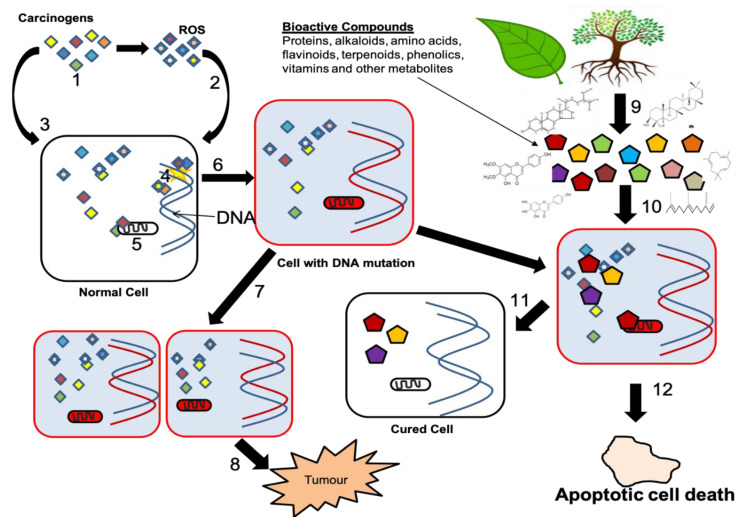
Carcinogenesis and the mechanism of action through which plant-derived bioactive compounds induce cell death.

**Figure 3 molecules-25-04435-f003:**
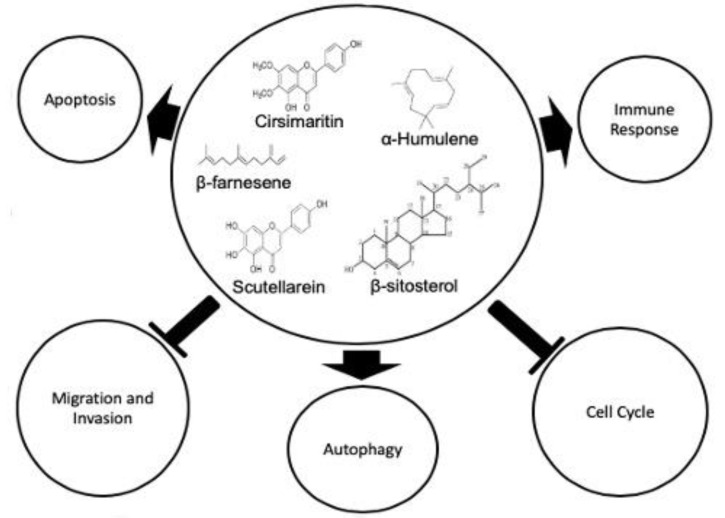
Possible mechanisms of action induced by plant-derived phytochemicals.

**Figure 4 molecules-25-04435-f004:**
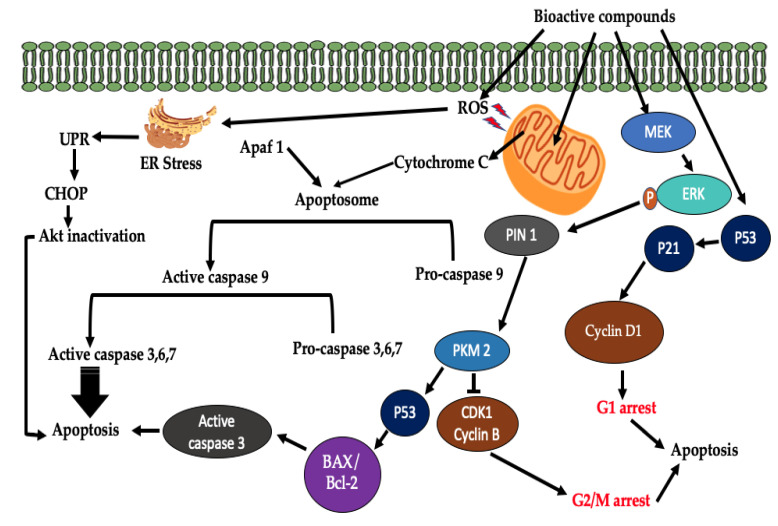
General cell death signaling pathways induced by plant-derived bioactive compounds.

**Table 1 molecules-25-04435-t001:** Shows the general taxonomy and common *Dicoma* species.

Kingdom	Phylum	Class	Order	Family	Genus	Species
Plantae	Magnoliophyta	Magnoliopsida	Asterale	Asteraceae	*Dicoma*	*anomala* *arenaria* *capensis* *fruticosa* *galpinii* *kurumanii* *macrocephala* *montana* *picta* *prostrata* *schinzii* *swazilandica* *tomentosa*

**Table 2 molecules-25-04435-t002:** The medicinal uses of *D. anomala* Sond.

Habitat	Part(s) Used	Medicinal Use	Country Practiced
Stony grasslands, rocky hillsides, and savanna forests. Altitude from 165 to 2075 meters.	Roots	Abdominal Pain	Zimbabwe
	Roots	Cardiovascular Diseases	Namibia and South Africa
	Tuber	Asthma	South Africa
	Leaves and roots	Breast Cancer	Lesotho
	Roots	Cataracts	Zimbabwe
	Leaves and roots	Diabetes	Lesotho and South Africa
	Root	Renal Problems	Swaziland and South Africa
	Root	Malaria	Zimbabwe
	Root	Pneumonia	South Africa and Zimbabwe
	Root	Syphilis	Zimbabwe
	Root	Cough	Malawi, Namibia, and South Africa
	Root	Hemorrhoids	Namibia
	Root	Intestinal warms	Lesotho, Malawi, and South Africa
	Flower and roots	Wounds and sores	Lesotho, Malawi, and South Africa

**Table 3 molecules-25-04435-t003:** Shows phytochemical compounds and their role in cancer therapies.

Plant Name	Phytochemicals	Role in Cancer Therapy	Reference
*Nigella sativa*	Thymoquinone	Targets the signal transducer and activator of transcription factor 3 (STAT3) pathway thereby leading to the inhibition of cancer cell proliferation.	[20]
*Petroselinum crispum*	Apigenin	Targets intrinsic apoptotic pathways. In lung cancer, apigenin exert its effects by modulating signals between Akt and Snail/Slung signaling pathways leading to metastatic restrain of cancer cells.	[21]
*Zingiber officianale*	6-Shogaol	Targets Akt and signal transducer and activator of transcription (STAT) signaling pathways. In NSCLC, 6-Shogaol directly regulates Akt1/2 pathways, which will in turn lead to the growth inhibition or induce apoptotic cell death.	[22]
*Thymus vulgaris*	Thymol	Targets the mitochondria and its effects induce mitochondrial malfunction and apoptosis of cancer cells.	[23]
*Scutellaria baicalensis*	Baicalein	Targets mitogen-activated protein kinase (MARPK), extracellular signal-regulated kinase (ERK), and p38 signaling pathways. In colon cancer, Baicalin induces apoptosis and growth suppression.	[24,25]
*Glycyrrhiza glabra*	Glycyrrhizin	Targets thromboxane A2 (TxA2) and signal transducer and activator of transcription (STAT) pathways.	[26]
*Oldenlandia diffusa*	Ursolic acid	Targets and interferes with cancer protein Ki-67, CD31, and microRNA 29 (miR-29a).	[27]
*Melilotus officinalis*	Dicumarol	Targets pyruvate dehydrogenase kinase 1 (PDK1) leading to the interference of the intrinsic apoptotic pathway	[28]
*Glycyrrhiza glabra*	Licochalcone A	Targets cyclins and cyclin-dependent kinases (CDKs). Their interaction with the cyclins and CDKs results in cell cycle arrest in the G_0_ or G_1_ and G_2_ or Mitotic phases.	[29]

**Table 4 molecules-25-04435-t004:** The structure of major plant-derived anticancer compounds.

Anticancer Agent	Structure	Classification	Reference
α-Humulene	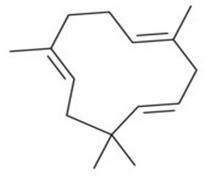	Sesquiterpenes	[50]
β-farnesene	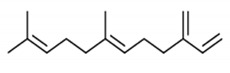	Sesquiterpenes	[51]
Cirsimaritin	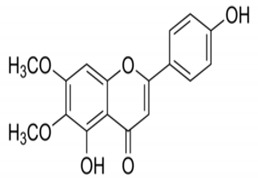	Flavonoids	[52]
Scutellarein	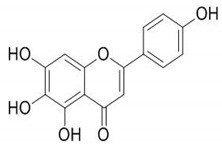	Flavonoids	[53]
β-sitosterol	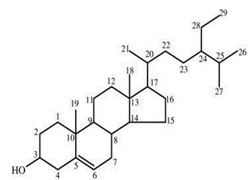	Phytosterols	[54]
Stigmasterol	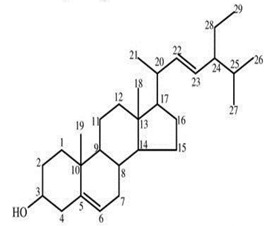	Phytosterols	[54]
Taraxasterol	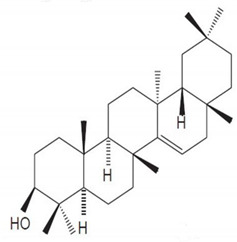	Phytosterols	[55]
Lupeol	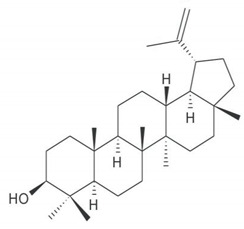	Triterpenes	[56]
Lupenone	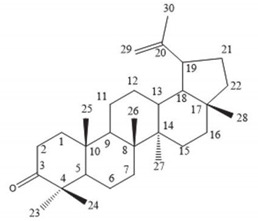	Triterpenes	[57]

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
