# Peer review of "Potential Treatment of Breast and Lung Cancer Using Dicoma anomala, an African Medicinal Plant"

_molecules, 2020, doi:10.3390/molecules25194435_

Round 1

Reviewer 1 Report

The manuscript is well written and pleasant to read.

In my opinion two important parts are missing for the publication:

-the role of Dicoma extracts on molecular target as shown in figure 3. The authors should describe the effects of Scutellarein, Cisrsimaritin…..et al. On apopotosis, migration and cell cycle.

- Cancer stem cells, in particular breast cancer stem cells, are very important to targeting cancer:

the authors should describe the effects of Dicoma extracts on Cancer stem cells to improve Manuscript

Author Response

Reviewer 1 Comments 

Authors Response

The role of Dicoma extracts on molecular target as shown in figure 3. The authors should describe the effects of Scutellarein, Cisrsimaritin…..et al. On apopotosis, migration and cell cycle.

Authors thank the reviewers for the comments.

The comments were corrected according to reviewers’ recommendations (page 6-7, line 126-172).

Cancer stem cells, in particular breast cancer stem cells, are very important to targeting cancer:

Corrected and changed accordingly (page 12 line 239-242).

The authors should describe the effects of Dicoma extracts on Cancer stem cells to improve Manuscript.

Corrected and changed accordingly (page 12 line 235-242).

Reviewer 2 Report

The manuscript seems to be interesting. There are some minor corrections which are crucial for the acceptance of the current manuscript. 

  1. The abstract is focused on global cancer types while the title is only focused on two major cancer types. It would be interesting and highly suggested that authors should expand the landscape of the current manuscript by including other cancer types. 
  2. Authors should expand and provide an easy descriptive figure that depict the mechanism(s) responsible for the potential anti-cancer activity of the D. anomala. 
  3. The manuscript is missing the future perspectives section, that is crucial for understanding what gaps are needed to be addressed for future research on this plant.
  4. English language needs some attention. 

Author Response

Reviewer 2 Comments 

Authors Response

The abstract is focused on global cancer types while the title is only focused on two major cancer types. It would be interesting and highly suggested that authors should expand the landscape of the current manuscript by including other cancer types. 

Authors thank the reviewers for the comments.

The comments were corrected and expanded according to reviewers’ suggestions (page 13-14 line 310-328)

Authors should expand and provide an easy descriptive figure that depict the mechanism(s) responsible for the potential anti-cancer activity of the D. anomala

The comments were corrected according to reviewers’ recommendations (see figure 4, page 8, line 177-178).

The manuscript is missing the future perspectives section, that is crucial for understanding what gaps are needed to be addressed for future research on this plant.

The comments were carried out according to reviewers’ suggestions (page 14-15, line 349-366).

Round 2

Reviewer 1 Report

It is ok now